# *“I Thought It Was Better to Be Safe Than Sorry”:* Factors Influencing Parental Decisions on HPV and Other Adolescent Vaccinations for Students with Intellectual Disability and/or Autism in New South Wales, Australia

**DOI:** 10.3390/vaccines12080922

**Published:** 2024-08-16

**Authors:** Allison Carter, Christiane Klinner, Alexandra Young, Iva Strnadová, Horas Wong, Cassandra Vujovich-Dunn, Christy E. Newman, Cristyn Davies, S. Rachel Skinner, Margie Danchin, Sarah Hynes, Rebecca Guy

**Affiliations:** 1Kirby Institute, Faculty of Medicine and Health, UNSW Sydney, Sydney, NSW 2052, Australia; cklinner@kirby.unsw.edu.au (C.K.); ayoung@kirby.unsw.edu.au (A.Y.); rguy@kirby.unsw.edu.au (R.G.); 2Australian Human Rights Institute, UNSW Sydney, Sydney, NSW 2052, Australia; 3Faculty of Health Sciences, Simon Fraser University, Burnaby, BC V5A 1S6, Canada; 4School of Education, UNSW Sydney, Sydney, NSW 2052, Australia; i.strnadova@unsw.edu.au; 5Disability Innovation Institute, UNSW Sydney, Sydney, NSW 2052, Australia; 6Susan Wakil School of Nursing and Midwifery, University of Sydney, Sydney, NSW 2050, Australia; horas.wong@sydney.edu.au; 7Centre for Social Research in Health, UNSW Sydney, Sydney, NSW 2052, Australia; c.newman@unsw.edu.au; 8School of Public Health, University of Queensland, Brisbane, QLD 4072, Australia; c.vujovichdunn@uq.edu.au; 9Specialty of Child and Adolescent Health, Faculty of Medicine and Health, University of Sydney, Sydney, NSW 2050, Australia; cristyn.davies@sydney.edu.au (C.D.); rachel.skinner@health.nsw.gov.au (S.R.S.); 10Sydney Infectious Diseases Institute, University of Sydney, Sydney, NSW 2050, Australia; 11Murdoch Children’s Research Institute, Melbourne, VIC 3052, Australia; margie.danchin@rch.org.au; 12Department of Paediatrics, University of Melbourne, Melbourne, VIC 3052, Australia; 13Health Protection NSW, Sydney, NSW 2065, Australia; sarah.hynes@health.nsw.gov.au

**Keywords:** HPV, vaccine access, vaccine hesitancy, parents, adolescents, intellectual disability, autism, special schools, school staff, immunisation providers

## Abstract

The uptake of human papilloma virus (HPV) and other adolescent vaccinations in special schools for young people with disability is significantly lower than in mainstream settings. This study explored the factors believed to influence parental decision making regarding vaccine uptake for students with intellectual disability and/or on the autism spectrum attending special schools in New South Wales, Australia, from the perspective of all stakeholders involved in the program. Focus groups and interviews were conducted with 40 participants, including parents, school staff, and immunisation providers. The thematic analysis identified two themes: (1) appreciating diverse parental attitudes towards vaccination and (2) educating parents and managing vaccination questions and concerns. While most parents were described as pro-vaccination, others were anti-vaccination or vaccination-hesitant, articulating a marked protectiveness regarding their child’s health. Reasons for vaccine hesitancy included beliefs that vaccines cause autism, concerns that the vaccination may be traumatic for the child, vaccination fatigue following COVID-19, and assumptions that children with disability will not be sexually active. Special school staff regarded the vaccination information pack as inadequate for families, and nurses described limited educational impact resulting from minimal direct communication with parents. More effective communication strategies are needed to address vaccine hesitancy among parents with children with disability.

## 1. Introduction

Vaccination coverage in young people with disability remains suboptimal in Australia and globally [1]. Although school-based vaccination programs are one of the most effective ways to reach young people [2,3], four Australian studies found a 53 to 66% uptake of adolescent vaccines among students with disability in special schools [4,5,6,7], which is significantly lower than the rates of roughly 73 to 82% for the general school population [8,9]. Evidence from the UK and the US has identified similar inequities, especially among adolescents with intellectual disability and/or on the autism spectrum [10,11,12,13,14,15]. Understanding and addressing lower vaccination rates in these populations is vital. Studies show individuals with disability face increased risks of vaccine-preventable diseases, hospitalisations, and complications due to underlying social determinants [16,17]. Regarding HPV-associated diseases in particular, research suggests that women with disability have higher odds of late-stage cervical cancer [18,19].

Limited research indicates that parental consent and vaccine hesitancy may contribute to reduced coverage among young people with disability, among other critical barriers such as inadequate management of needle anxiety and lack of guidelines about reasonable adjustments [14,20]. The World Health Organisation has recognised vaccine hesitancy as a major global health concern [21]. It has been described as “a motivational state of being conflicted about or opposed” to vaccines [22]. Compared to the 3 to 7% of parents who outright refuse vaccination, about one-fifth of parents are described as hesitant about vaccines [23,24,25,26,27], but flexible in their views if provided with information tailored to their social and personal contexts [28]. However, very little research has examined vaccine hesitancy among parents of adolescents with disability. Some studies suggest that vaccine decision making in this group is influenced by factors beyond just safety concerns, and include issues such as beliefs that vaccines cause autism, fears that the procedure may cause anxiety or trauma, and, in relation to HPV vaccination, perceptions that their child will not be sexually active [10,13,14,15,29,30], despite evidence to the contrary [31]. However, these studies have relied on surveys or chart reviews and in-depth qualitative data exploring the issue within a socio-ecological environment is lacking [32,33,34].

Strategies to increase vaccination in young people with a disability are of public health importance as adolescents with disability make up a 10% of the adolescent population globally, or about 240 million [35,36]. In Australia, 10% (or 380,000) of school students have disability [37], with 5.4% (or 206,000) having profound support needs [37]. Of students with disability, 65% (or 148,000) of males have intellectual disability and 40% (or 91,000) have psychosocial disability (defined as emotional, mental, and social or behavioural difficulties); this is compared to 54% (or 84,000) and 38% (or 58,000) of females, respectively [37]. Additionally, around 164,000 Australians have autism, with 83% under the age of 25 years [38]. Those with the highest support needs are more likely to attend special schools [37]. Despite their high support needs, there are few studies to understand vaccination coverage gaps in this population and no school-based interventions [39].

This study explored the factors believed to influence parental decision making regarding the uptake of HPV and other adolescent vaccines for students with intellectual disability and/or on the autism spectrum attending special schools in New South Wales, Australia. We included the perspectives of three stakeholder groups involved in the school-based vaccination program, including parents, school staff, and immunisation providers, to gain an in-depth understanding of the issue. Student interviews were also conducted but did not contribute data to this analysis.

## 2. Materials and Methods

### 2.1. Research Design and Setting

This study was part of Vax4Health (www.kirby.unsw.edu.au/vax4health, accessed on 7 May 2024), a four-year research project that aims to improve vaccination uptake and experiences in special schools in New South Wales, Australia. The project involves three phases: (1) Understand needs, (2) Co-design services, and (3) Pilot services. In this paper, we reported findings from Phase 1, which utilised a qualitative methodology guided by inclusive research principles [40,41]. This phase involved understanding why the uptake of adolescent vaccines is lower in special schools. These vaccines include diphtheria–tetanus–acellular pertussis (dTpa), human papillomavirus (HPV), and meningococcal ACWY (MenACWY), which are offered free at school, as part of the National Immunisation Program in Australia. Parental consent is required for each vaccine, and NSW Health works in partnership with schools to deliver the program in all school settings using local public health unit teams. In this study, fieldwork was conducted in four special schools, focusing on the perspectives of stakeholders who use (parents and students), deliver (health staff), and facilitate (education staff) the program. Special schools cater to students with intellectual disability, mental health issues, autism, physical disability, sensory impairment, learning difficulties, and behaviour conditions. The schools were purposively selected based on different demographics, such as socioeconomic factors, location, and size, with two schools being located in metropolitan areas and two in regional areas of NSW.

### 2.2. Participants and Recruitment

Using purposive and convenience sampling [42], we recruited 50 participants, including 10 students with intellectual disability and/or on the autism spectrum, 6 parents of adolescent children with disability, 24 education staff (teachers, school support staff, and education policymakers), and 10 health staff (immunisation nurses, vaccination program managers, and health policymakers). Data from adult stakeholder groups were included in the current analysis. Eligibility criteria for adult participants included the ability to verbally communicate in English and being over 18 years of age. The recruitment process was conducted through participating schools and local public health units. Recruitment invitations were sent to the principals, who then distributed information throughout the school community, inviting teachers, support staff, and parents to participate in this study.

### 2.3. Data Collection

Semi-structured interviews and focus groups were conducted until sufficient depth and diversity in the accounts were achieved [43]. In length, they were in the range of 30–60 min for adults and 10–20 min for students and were completed in person and online. Semi-structured facilitation guides were utilised, allowing for the exploration of new issues mentioned by participants. Questions included demographics measures as well as open-ended questions within three key realms: knowledge, attitudes, and experiences regarding the school vaccination program in special schools, knowledge and attitudes regarding the vaccines on offer, and ideas on potential interventions or areas for improvement. All sessions were audio-recorded and research staff took notes to record observations of critical moments and reflections.

### 2.4. Data Analysis

Data were thematically analysed [44] using a 7-step framework method for the analysis [45] and a team-based approach [46]. The first step involved professional transcription and de-identification of the data. The second and third steps involved familiarisation and initial coding (e.g., “parent is hesitant to get their child vaccinated due to concerns about side effects”), led by three authors (AC, AY, and CK). A subset of coded transcripts for each stakeholder group was then discussed and recurring codes identified in a participatory workshop involving five authors (AC, AY, CK, IS, and HW). In the fourth step, AC, AY, and CK grouped codes into categories and developed a draft analytical framework that was applied to the entire dataset. In step five, the coded transcripts were indexed to this framework in NVivo 12 [47], iteratively refining the framework along the way. Some datasets were indexed independently by two researchers (AC and CK) to ensure intercoder reliability. The final two steps (led by AC, AY, and CK) involved summarising specific patterns in the data for theme generation [44] and, ultimately, data interpretation and report writing. This phase was an ongoing process that involved reviewing all collated extracts for each theme and sub-theme, conducting and writing a detailed analysis, and organising the data into a coherent story, with theme names ascribed in the final analysis and write-up. Findings were then shared with the wider multi-disciplinary team at key project milestones for critical review and input. We believe this detailed, reflexive, multi-disciplinary team-based approach produced insightful and trustworthy findings.

## 3. Results

### 3.1. Participants’ Characteristics and Key Themes

Data from all adult stakeholders contributed to this analysis. Parents were all female, aged from 40 to 52 years, with three living in metropolitan areas and three in regional settings. School staff participants included 8 executive school staff (principals/deputy principals), 13 teachers and learning support officers, 3 administrative school staff, and 1 policy manager. Health staff participants included six nurse immunisers and team leaders, four program coordinators, and one policy manager. Based on participants’ narratives (either personal or witnessed), we identified two key themes related to the factors influencing parental positions on HPV and other adolescent vaccinations: (1) appreciating diverse parental attitudes towards vaccination and (2) educating parents and managing vaccination questions and concerns, each with three sub-themes. Key text in participant quotes was bolded by the authors to facilitate reading.

### 3.2. Appreciating Diverse Parental Attitudes towards Vaccination

According to our data, parents were seen to fall into one of three categories: (1) pro-vaccination without persuasion, (2) anti-vaccination and unlikely to change their minds, and (3) vaccination-hesitant and needing information personalised to their concerns. Reasons for vaccine acceptance, refusal, or hesitation are discussed within each sub-theme.

#### 3.2.1. Pro-Vaccination without Persuasion

Most parents in special schools were described as pro-vaccination, called “*believers*” by nurses. These parents believed in the value and benefit of vaccination without the need for convincing. Their attitudes stemmed from being well-educated in health, having had a vaccine-preventable illness, or conforming to social expectations or health guidelines.


*“I think for me, it’s more because they recommend it. I am kind of very obedient patient, whatever I get (laughs). We say, if we are told the Government recommends, we try it.”*

*Parent (P3)*


School staff observed that parents consented to the vaccines to protect their already vulnerable children from preventable illnesses.


*[Q Why do you think parents make the decision to have their children vaccinated?] “I think it’s because they do want to protect them… our kids are very vulnerable and the parents are very aware of trying to keep them safe and healthy.”*

*School staff (SS3)*


For the HPV vaccine specifically, parents who supported it were either well-informed about its efficacy against cancer, followed official health advice, or chose “*to err on the side of caution*” even if they thought their child might not need it.


*[Q Did he receive all the adolescent vaccinations? I think it’s Meningococcal, DTPA and the HPV vaccine.] Ah, I think so, I would actually have to consult the records to be a hundred percent sure. I definitely remember him getting the HPV one, cos I remember thinking “oh, I don’t’ know if that one’s necessary to him in his context”. Whether or not he’s going to grow up to have a normal, you know, sexual life when he’s an adult, I don’t know, because of his disability. But I thought it was better to be safe than sorry.”*

*Parent (P4)*


#### 3.2.2. Anti-Vaccination and Unlikely to Change Their Minds

A “*small minority*” of parents in special schools were described as anti-vaccination or “*objectors*” by nurse and school staff participants. These parents were said to be against vaccination, from the outset, and difficult to convert in their attitude. This cohort was thought to have grown in size since COVID-19.


*“We probably got two families in the school I guess that are anti-vaxxers.” [Q How do you know?] “Because they tell me. They’ll say, ‘Absolutely no way I want my child vaccinated. They’re not to be vaccinated.’”*

*School staff (SS2)*



*Then we receive the emails or the phone calls you know, “do not vaccinate my child under any circumstances, or this will end up in a lawsuit…” We’ve had quite a few of those. More so since the Covid sort of thing went down.*

*Immunisation staff (IS9)*


Pro-vaccination parent participants described anti-vaccination parents’ views as stemming from misinformation and political rhetoric, often retrieved from social media sources, particularly regarding lack of trust of “mainstream medicine.”


*“People get really narky, get bogged down in their beliefs and don’t sort of put the blinkers up and don’t want to hear alternative theories…If you give them the right information…some people are very conspiracy theory minded and will just go ‘that’s just’ you know, mainstream medicine trying to convince you, but we know the truth.’”*

*Parent (P2)*


Several participants also attributed vaccine rejection to the belief that their child’s childhood vaccinations had caused their autism. This was thought to lead some parents to refuse further vaccinations to not make their child’s autism worse.


*“I had a conversation with two other parents, and they were both absolutely adamant that that’s why their children have autism. They were, in their words, “normal”, had their vaccinations and then very soon after started displaying Autistic behaviours. And even though there’s so much evidence that that’s not the case, but when that’s what you see, you know, you’ve got a child and you have seen a difference after a vaccination, it doesn’t matter how much research they read, if that’s what they believe, that’s what they believe.”*

*School staff (SS6)*


Overall, anti-vaccination parents were seen by health and school staff participants as having “*very strong personalities*” and were met with respect and provision of factual information but no persuasion. Pro-vaccination parents echoed nurses’ sentiment that anti-vaccination parents were hard if not impossible to convert through education.


*“I don’t think providing information is gonna change those people’s minds… It’s really hard to reason with people like that because they’re not being reasonable.”*

*Parent (P4)*


#### 3.2.3. Vaccination-Hesitant and Needing Information Personalised to Their Concerns

The final group of parents were described as vaccination-hesitant, or “*fence-sitters*” by nurses due to their uncertainty about vaccines. Generally, special school parents were seen by teaching and nursing staff participants as more worried about the potential harm from vaccination than parents of adolescents without disability. This heightened sensitivity was attributed to their child’s complex health needs and medical history, leading some parents to forego school vaccination if they could not see a guaranteed advantage.


*“I tend to find that [special school] parents have a lot of anxieties and concerns because they’ve been within that system for a long time, particularly health systems… They’re very well informed in a lot of aspects. And I think that they really try to make these decisions with the best of intentions… What I’m trying to say, for a child without a disability, their parents probably wouldn’t exactly think twice sometimes. They’d probably just go “yep, sure, no worries, cos you’re healthy.” Whereas these parents I guess, it’s almost like that helicopter, that over-protectiveness. Like, they kind of really want to make sure that what they’re doing is gonna offer them an advantage, given that there’s already so many things that they’re having to deal with.”*

*Immunisation staff (IS2)*


Parents’ vaccination concerns centred around adverse health effects for their child, as well as worries about how their child would cope emotionally with the procedure including the potential use of restrictive practices. Consequently, they approached school vaccinations more cautiously, giving consent only when confident it would provide clear benefits.


*“They don’t want to do any harm to their child, especially with special needs children that already have quite a lot of problems… They don’t want to provide any more harm to their child. But they also want to protect their child from infection as well… You know, ‘can you guarantee this won’t harm my child?’ … is the main thing.”*

*Immunisation staff (IS8)*



*“I think [teachers’ worries about the restrictive practice policy] was also kind of carried across in the parents who usually go ‘I’ll consent but we might give them a go when they’re getting dental work done at the children’s hospital or something.’ … So I think there’s some anxieties about how the students might react at clinic, that kind of prevents them from consenting.”*

*Immunisation staff (IS8)*


Some parents could not see a clear benefit in relation to the HPV vaccine because they did not expect their child to become sexually active due to their disability.


*“I said no to the HPV one because, as hard as this, I don’t believe they’re gonna be sexually active. So, for me that would be the reason if they were going to be sexually active, that’s yeah, so I didn’t um, say yes to that one.”*

*Parent (P3)*


The intersection of disability with gender and age was also evident in HPV-hesitant attitudes, with nurse participants noting that some parents thought that the vaccine was not necessary for boys or that their child was “*too young*” to receive it.


*“Sometimes the boys aren’t consented for the HPV vaccine because I guess parents think that boys aren’t gonna get it cos it’s related to cervical cancer.”*

*Immunisation staff (IS5)*



*“It being a sexually transmitted disease that we’re looking to prevent, a lot of parents don’t see the necessity at that time, for those students in Year 7… they’re already saying ‘oh, but my kid’s not gonna be sexually active for another, you know, forty years,’ they think, ‘why do I need to vaccinate them now?’”*

*Immunisation staff (IS2)*


Education and health staff participants also reported noticing carer burnout and COVID-19-induced vaccination fatigue and confusion around the number of vaccines needed.


*I think some parents have got that idea of “my kid’s had enough vaccinations in the last two years, I don’t really want any more vaccinations at the moment”*

*Immunisation staff (IS6)*



*“Some of them are burnt out as well. Amongst everything that their children require. There’s a lot of (inaudible) to the doctor, or fight with the kid over the doctor…”*

*School staff (SS9)*


Teacher participants believed that some parents “*just don’t care*” or “*don’t know the consequences*” of non-vaccination, emphasising the need for more detailed education around the importance of vaccinations. Stakeholders agreed that vaccination-hesitant parents could be swayed by tailored, easy-to-understand information that addressed their specific concerns.


*“if they’re hesitant, yes, providing the facts in an easy to understand way, might overcome their hesitancy…”*

*Parent (P4)*


### 3.3. Educating Parents and Managing Vaccination Questions and Concerns

In participants’ discussions of the vaccine information available to parents, we identified three sub-themes: (1) Perceptions of the government vaccination information pack; (2) Role of schools in informing parents about the vaccination program; and (3) Role of nurses in shaping parental vaccination decisions.

#### 3.3.1. Perceptions of the Government Vaccination Information Pack

Health staff participants noted that one standard government vaccination information pack is delivered to all school types. They described this information pack as “*quite a detailed form that [parents] need to read in order to provide informed consent*”. When asked if anything was tailored to the needs of families with disability, a school program coordinator within a public health unit indicated that “*all the information is the same*”.

Education staff participants considered this information as inadequate to cater for the needs of families of children attending special schools. Reasons cited included low education levels, limited health literacy, and the “*special needs*” of “*a lot of*” families, along with their heightened sensitivity to their child’s complex medical needs.


*“I just wonder to what extent these packs are accessible. Like especially if parents themselves might have intellectual disability...”*

*School staff (SS2)*



*“We have a large, and I mean I’m talking around seventy percent low socio-economic, have not finished year ten. So, realistically, they can’t even read the forms.”*

*School staff (SS4)*


A health staff participant responsible for managing the school vaccination program indeed reported of regularly receiving inquiries about whether the program was available to students with a disability. A nursing team leader also suggested that, as the state transitions from paper to online consent (initiated in 2023), there could be an opportunity to customise the information pack for families of special school students.


*“Perhaps there is a gap in the program in where we need a leaflet, or brochure, for parents of children attending special schools…and perhaps part of that [online] process we can include advice…”*

*Immunisation staff (IS10)*


Health staff participants suggested improving vaccine education by making the information pack “*disability-friendly*” and easy to read for low-literacy parents. Suggestions included addressing common questions and concerns of parents of adolescents with disability, such as dispelling the vaccines-cause-autism myth and providing reassurance on how nursing teams will support young people during vaccination if they are nervous about needles.


*[Q What can be done to improve vaccination coverage in special schools?] I think potentially making the resources a little bit more explicit, cos it’s very general, … potentially that’s something we need to look at internally and developing resources that kind of address these questions that parents might have. Like, “My child has autism, can they still be vaccinated at school? What can you do for them at school if they have particular, you know, sensitivities or that kind of thing…” Or could we be making resources that are disability friendly…or easy to read… Or even for students to have a look at, in terms of educating them about why this is happening…*

*Immunisation staff (IS3)*


#### 3.3.2. Role of Schools in Informing Parents about the Vaccination Program

The government information pack is provided to parents via the schools, including contact details of the public health units if parents wish to discuss vaccine-related questions. Thus, while schools were the first point of contact for parents, health staff believed that schools generally refrained from providing medical advice.


*“Most parents feel a lot better about emailing the school…if they do have any queries or concerns... ‘I don’t want to get my child vaccinated, or I do want to get my child vaccinated, how do I go about it?’ And then the [school] coordinator will either answer their question or forward it onto us. In terms of general communication about vaccines and their importance and ‘this is why you should get it’, the school coordinators don’t generally do that. It depends on kind of their background.”*

*Immunisation staff (IS3)*


Several school staff participants echoed this sentiment. When approached by parents, they generally wanted to remain neutral, citing their lack of health expertise, and referring parents to local public health units as the authorities. As one participant said, they “*leave it between the parent and Health*”.


*“Especially as we’re not a medical professional, I don’t think we can legally say ‘we think your child can get this’, because if something went wrong. So I guess if a parent had questions, I would try to refer them to somewhere to get the right answers but I would never recommend or I would not tell a parent they should be doing this…”*

*School staff (SS6)*


One teacher participant, however, openly admitted to influencing parents to get their children vaccinated by sharing their personal pro-vaccination views and aiming to ensure all students were vaccinated since they “*had so many vulnerable students*”. However, school staff participants also emphasised their respect for parents’ ultimate vaccination decisions.


*“We do try and say you know, why we think these things are a good idea but I can’t make that choice for them. […] it’s completely the parental choice.”*

*School staff (SS3)*


Some teachers saw a more formal role for them to inform the school community about vaccines as part of their educational mandate but they emphasised doing so without an “*agenda*”, allowing parents to make their own decisions. In such instances, vaccination information would be simplified and referenced to medical evidence.


*I would simplify it. I would say the Human Papilloma[virus], “this will prevent cancer”, and…the reason we give it to our students, is because there is evidence to say…*

*School staff (SS4)*


Some teachers, however, were observed by nurse participants to *incidentally* convey misinformation to parents, particularly concerning the HPV vaccine. These tended to involve situational side comments, possibly born out of teachers’ desire to make things easier for parents.


*“I think HPV, there’s still a little bit of that “oh, it’s a girls’ vaccine” type and I sometimes hear teachers saying that to parents. It’s for girls, you don’t have to get your son that HPV vaccine.”*

*Immunisation staff (IS3)*


Despite the inaccuracy of such comments, there was a shared attitude of care among school staff participants for the wellbeing of their families, as exemplified by the following remark from a school principal.


*[Q It sounds like you’re very positive in terms of trying to give them that information in order to make that decision] “Yeah. I think it’s more around the questions around the vaccination, like will it hurt, will they get sick, what are the side effects, the consequences. And to reinforce that we will monitor your child, we will make sure that they’re ok.”*

*School staff (SS7)*


#### 3.3.3. Role of Nurses in Shaping Parental Attitudes to Vaccination

Despite being seen as “*a trusted authority*”, nurse participants pointed out how limited their educational impact on parents was, as they usually had no or very little personal and only isolated one-off phone contact with parents. Therefore, there was little opportunity to become acquainted with parents and understand what kind of education they needed.


*“We’re not even really interacting with the parents.”*

*Immunisation staff (IS7)*



*“We don’t have any way of communicating with parents directly.”*

*Immunisation staff (IS3)*



*“If parents don’t provide consent, we can’t contact them, unless they contact us.”*

*Immunisation staff (IS8)*


Nurse participants noted that they typically dealt with a “*small percentage*” of parents, and, in most cases, these were individuals who already wanted their children vaccinated but had some questions.


*“The people who are ringing me aren’t vaccine hesitant, they’re people wanting to genuinely get their kids vaccinated.”*

*Immunisation staff (IS9)*


To help parents reach out to them, some nurses provided their personal contact details on the vaccination packs for parents to reach out to them with questions or concerns. Others maintained social media pages or held parent information sessions. Education encompassed factual information, websites for further information and, occasionally, referral to immunisation specialist centres. Regarding the HPV vaccine, nurse participants tended to de-emphasise the sexual aspect of it so as not to “*put parents on alert*”.

Importantly, nurses acknowledged the challenge of reaching silent parents, i.e., those with vaccination questions or concerns “*that aren’t turning up*”. They also noted that, for education to be effective, they needed to become acquainted with parents and tailor their information to parents’ specific needs and concerns.


*“I’ve gotta find out what their concerns are and then tailor my response to that... So, it’s about good communication and listening as well, and finding out what is it that they’re worried about? … and you know, we talk to them about weighing the risk of side effects of the vaccination and providing them with how the vaccine does work, what it’s protecting the child against…”*

*Immunisation staff (IS8), nurse*


One nurse participant emphasised the importance of meeting parents’ needs more holistically, recognising them as individuals with rich histories, social contexts, and personal needs.


*“It’s about the information that we’re actually putting out there to them. And I think it’s also about understanding the individual and all of the individual circumstances…We really probably need to be able to see the entire picture to go “oh, yeah, alright…if these are your concerns or this is your history, or this is what you’ve experienced, this is the information that you need to know about.”*

*Immunisation staff (IS2), nurse*


## 4. Discussion

### 4.1. Results in Context

This study revealed the complex and diverse nature of vaccination decisions in parents of adolescents with intellectual disability and/or on the autism spectrum, from the perspective of stakeholders involved in the vaccination program in special schools. On one end of the spectrum were parents who accepted vaccines from the outset based on their education or adherence to health guidelines, without the need for convincing. At the other end of the spectrum were parents opposed to vaccines due to misinformation, mistrust of the government, and misbeliefs that vaccines can cause autism. Toward the middle of the spectrum were parents who were hesitant to vaccinate their child due to a variety of unaddressed questions and concerns, including fears of adverse health effects, worries about the use of restrictive practices, concerns over how their child would cope with the procedure, caregiver burnout, vaccination fatigue and confusion, and a false assumption that people with disability do not require sexual health prevention. We also identified gaps in the provision of vaccination information, such as a lack of accessible informed consent processes for families of special school students and weak communication channels between nurses and parents. These results have important implications for developing more effective vaccine communication strategies to ensure parental consent and provision of information is accessible, inclusive, and tailored to families’ individual needs.

Our finding that parents strongly opposed to vaccination are believed to be hard to assure of the importance of vaccinations is supported by previous research [48,49]. As scholars of disinformation posit, vaccination deniers are extremely hard to convince of the life-saving value of vaccines [50,51] and trying to persuade them, even with factual information, can sometimes strengthen their beliefs [52]. Our data indicate that fears linking vaccines to autism remains a highly emotive issue among some families with disability, making education difficult. Our analysis also suggests that the COVID-19 pandemic exacerbated fatigue and confusion of vaccines in this population, a trend consistent with other studies [53]. Importantly, however, our study corroborates existing research [23] showing that strongly held anti-vaccination views represent only a small minority of families in Australia. More parents in special schools were believed to be vaccination-hesitant, with legitimate but unaddressed questions and concerns about vaccines and the vaccination procedure [23]. It is this group “for whom real gains can be made” ([23] p. 445) with targeted communication strategies that are tailored to families of adolescents with disability and high support needs. Such strategies need to aim at both increasing awareness of disease risks and correcting vaccination misbeliefs [23,54].

A key finding and common thread throughout interviews was that the information pack for parents was not tailored to the needs of families with adolescents attending special schools due to two issues in these communities: low health literacy and marked sensitivities regarding the impact of vaccines and vaccinations on the wellbeing of their child. In the first regard, participants’ accounts suggested that special schools are likely to cater for families with high levels of socio-economic disadvantage [55], impacting on their ability to understand the vaccination information and provide consent. Regarding parents’ sensitivities, one possible explanation for parents hesitating to get their child vaccinated at school might be the child’s needle anxiety, which may have built up over the course of their extended history of medical/needle-related procedures [56,57,58]. Indeed, children with intellectual disability and/or on the autism spectrum have been reported to have greater needle fear and poorer management of needle-related pain and anxiety [59]. These pain experiences are processed differently [60] and can causes distress and behavioural challenges [61]. Known needle anxiety may contribute to vaccine hestinacy in some parents [62,63], especially if this coincides with not being convinced that the vaccination benefits will outweigh the risk of experiencing vaccination-related trauma [64].

Such benefits were not seen to outweigh risks in relation to the HPV vaccine, with some parents in this study expressing perceptions of low HPV-related disease risk on account that disability precludes sex. Our data also showed an avoidance by some nurses to discuss the sexual nature of the viral transmission in health education, likely due to lack of comfort in general about sexual health topics with adolescents with disability. This is consistent with a recent scoping review that shows many parents and health professionals do not see young people with intellectual disability as sexual objects, with agency and capacity for sexual relationships [31]. However, evidence suggests that there is a mismatch between parental beliefs about the sexual health of young people with intellectual disability and lived experiences, with national studies reporting similar rates of sexual intercourse [65,66], lower rates of contraceptive use [65,66,67], and higher rates of STIs [67,68] compared with those without disability. This population also experiences high rates of sexual abuse [69] and low cervical cancer screening uptake [70]. Thus, HPV vaccine uptake is vital. However, it seems these unique concerns have been lost in the design of a one-size-fits-all school vaccination information strategy.

Another important finding in this study was a lack of clarity around who exactly is responsible for educating parents about vaccines and addressing their questions and concerns, which echoes previous research [71,72,73]. While schools serve as a conduit for communication about the school vaccination program between public health units and families, teacher participants did not view themselves as vaccination experts. They did not want to take responsibility of providing “wrong” advice to parents. Meanwhile, nurses lacked an official information pathway to individually liaise with parents who did not submit a vaccination consent form by a certain date, relying instead on parents to reach out to a public health unit. For most families, however, public health units are unfamiliar health institutions, which may make parents hesitant to come forward with their questions. This means that parents who hold misinformation, misconceptions, feelings of marginalisation, or unaddressed concerns may fall through the cracks because the information at hand, i.e., the standard school vaccination info pack, is not accessible nor sufficient to facilitate truly informed vaccination decisions. It is important we work towards making our health services more inclusive, and reasonable adjustments to the school vaccination consent process could be regarded as our duty of care under the Commonwealth Disability Discrimination Act 1994 [74,75].

### 4.2. Implications and Recommendations

Our results suggest a need for tailored communication designed to effectively address vaccine hesitancy in families with adolescents with intellectual disability and/or on the autism spectrum. Parents’ health literacy, vaccination (mis)beliefs, and specific concerns related to the disability of their child need to be considered. As previous research has found, parents of children with disability want healthcare that is “individualized, coordinated, easily accessible, and takes into account the entire family dynamic” [76].

Our first recommendation, based on participants’ improvement ideas, is to offer a disability-friendly information pack to all parents, in addition to the standard booklet. This pack could include multiple formats, such Plain English, Easy Read, and video or audio recordings, to provide more parents with access to information. In addition to presenting information on the risks of vaccines and vaccine-preventable diseases, we also recommend practical information and an FAQ section to ensure that typical questions, concerns, and misbeliefs of parents of children with disability are addressed [77]. Topics could include dispelling myths about vaccines and autism, promoting better HPV awareness, preparing students for vaccination, supporting children who are nervous about needles, and referral pathways for distraction and wake sedation in tertiary and other centres for students with extreme needle anxiety [78].

Our second recommendation, from the participants’ accounts, is to determine how immunisation teams can establish direct communication with parents who have not submitted a consent form by a specific date. This may require legislative changes as current legislation prevents the NSW Health Department from directly contacting parents in the context of the school vaccination program. However, educating hesitant parents about the importance of vaccinating their child is essential and should not be left at the discretion of schools. A two-way interaction with parents would allow nurses to identify and resolve areas of misunderstanding or concern, especially amongst parents with low health literacy and those with worries around their child’s needle anxiety, pain sensitivity, and/or need for the HPV vaccine. The education approach identified by nurse participants in our study points towards a person-centred model of care whereby parents and their children are met by healthcare professionals as individuals with unique histories and backgrounds, and with diverse values and beliefs shaped by their personal and social contexts [79,80]. Shared decision making between service providers and service users is another key component of person-centred practice. Interventions that facilitate this approach have been shown to reduce vaccine hesitancy and promote informed choice [81].

### 4.3. Strengths and Limitations

Limitations of this study include the small number of parent participants and the lack of gender and cultural diversity among the participants. All interviewees identified as female and White. In addition, we were only able to interview two parents who were opposed to certain vaccines, due to recruitment challenges with this group. Therefore, the dataset may not represent the full spectrum of views of parents with children with intellectual disability and/or on the autism spectrum. The difficulties of recruiting hard-to-reach, hidden, and vulnerable populations, including vaccine-hesitant parents, has been cited in recent research [82]. Findings suggest that challenges relate to the sensitivity of the topic, distrust of experts, and stigma towards people with anti-vaccination views [82]. Future research should develop more reflexive recruitment strategies that foster trust with communities to increase the diversity of samples [82].

Notwithstanding the limitations, this study incorporated a wide range of stakeholder viewpoints from 40 participants, adding rich, contextualised information based on personal experiences of the school vaccination program from the perspectives of all stakeholders involved in the program. Our choice of sample size follows common practice in qualitative research [83,84], where a small sample size enhances relationality and “fine-grained, in-depth inquiry in naturalistic settings” [85] (p. 483), central for the in-depth approach we took. In our study, 40 participants were a sufficient sample size to uncover the diversity and richness of perspectives and to reach data saturation.

Another strength of this study was that the coding and analysis of the data were completed by a diverse team of highly experienced researchers with interdisciplinary expertise in disability, special education, adolescent health, vaccination, social work, and qualitative research methodology. This encouraged deep discussions of the interview material and robust grappling with diverse perspectives, which helped to uncover potential researcher biases and enhance the validity, reliability, and credibility of our data interpretations [86].

## 5. Conclusions

This study highlights how the vaccine decision-making processes of parents of adolescents with intellectual disability and/or on the autism spectrum are complex and multifactorial. Vaccination hesitancy and lack of accessible, targeted information to facilitate informed consent point to undiscovered and unmet education needs among families at special schools, which are necessary to promote healthy outcomes in adolescents with disability. Addressing these issues may help to raise the low uptake of the HPV vaccine and other adolescent vaccines in special schools. These results will inform the next phase of the Vax4Health project to co-design vaccination communication materials for parents of adolescents with intellectual disability and/or autism. By tailoring to their sources of hesitancy, it may be possible to improve vaccine acceptance and uptake and ensure health equity.

## Data Availability

The data presented in this study are available upon request from the corresponding author. Due to ethical restrictions, they are not publicly available.

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
