# Peer review of "“I Thought It Was Better to Be Safe Than Sorry”: Factors Influencing Parental Decisions on HPV and Other Adolescent Vaccinations for Students with Intellectual Disability and/or Autism in New South Wales, Australia"

_vaccines, 2024, doi:10.3390/vaccines12080922_

Round 1

Reviewer 1 Report

Comments and Suggestions for Authors

Please see attached file for my review.

Comments on the Quality of English Language

Overall, the English language usage in the text is perfectly fine; a couple small error/issues (which caused me confusion) are mentioned in my review.

Reviewer 2 Report

Comments and Suggestions for Authors

In this mauscript, the authors explore the factors believed to influence parental decision-making regarding vaccine uptake for students with intellectual disability and/or on the autism spectrum attending special schools in New South Wales, Australia. They explore the perspective of all stakeholders involved in the program. While the manuscript is interesting, during the review process several concerns have raised:

1) Small sample size of only 50 participants, including only 6 parents of adolescents with disabilities. Potential risk of non-responden bias.
2) The study design was cross-sectional, capturing a snapshot of views and experiences at a single point in time. Longitudinal data would provide more robust insights into changes in attitudes and behaviors over time
3) The findings are specific to the context of special schools in New South Wales, Australia. This context-specific focus limits the ability to generalize the results to other regions or types of schools
4) Bias risk: lower number of opposed to certain vaccines (and this is indeed a key group) who could have offered a more balanced perspective
There are still myths and misconceptions of vaccines (no need of HPV vaccines due to low sexual activity, wrong believe about autism and vaccination)
5) The manuscript do not address unmet needs such as Who is responsible for educating parents about vaccines?

Round 2

Reviewer 2 Report

Comments and Suggestions for Authors

Endorsed